# Explosive fragmentation of Prince Rupert's drops leads to well-defined fragment sizes

Stefan Kooij [1✉], Gerard van Dalen[2], Jean-François Molinari [3] & Daniel Bonn[1]

Anyone who has ever broken a dish or a glass knows that the resulting fragments range from roughly the size of the object all the way down to indiscernibly small pieces: typical fragment size distributions of broken brittle materials follow a power law, and therefore lack a characteristic length scale. The origin of this power-law behavior is still unclear, especially why it is such an universal feature. Here we study the explosive fragmentation of glass Prince Rupert's drops, and uncover a fundamentally different breakup mechanism. The Prince Rupert's drops explode due to their large internal stresses resulting in an exponential fragment size distribution with a well-defined fragment size. We demonstrate that generically two distinct breakup processes exist, random and hierarchical, that allows us to fully explain why fragment size distributions are power-law in most cases but exponential in others. We show experimentally that one can even break the same material in different ways to obtain either random or hierarchical breakup, giving exponential and power-law distributed fragment sizes respectively. That a random breakup process leads to well-defined fragment sizes is surprising and is potentially useful to control fragmentation of brittle solids.

[1] Van der Waals-Zeeman Institute, University of Amsterdam, Amsterdam, The Netherlands. [2] Unilever Research and Development Vlaardingen, Olivier van Noortlaan, Vlaardingen, The Netherlands. [3] Department of Materials Science and Engineering, École Polytechnique Fédérale de Lausanne (EPFL), Lausanne, Switzerland. ✉email: s.a.kooij@uva.nl

Fragmentation is a process that plays a role in a great number of phenomena and applications, from tectonic plate motion to coffee grinding[1–12]. It covers length scales ranging from those of particle physics up to the size of the universe[13–15]. In many applications, control over the fragment size and fragment size distribution is of paramount importance. However, experiments have thus far shown that even if an object is broken in a fully controlled way, there is generally a large spread in the sizes of the resulting fragments. This spread in fragment sizes is a clear characteristic of power-law size distributions that are frequently reported in studies of brittle fragmentation[16–23], power laws, having only dimensionless fit parameters, contain no characteristic length scale, meaning they are scale invariant. This intriguing behavior has led, among other things, to the formulation of the concept of self-organized criticality[24], which suggests that it is impossible to control the fragment sizes, since they are given by the same underlying process for all fragmentation events.

We study here the fragmentation of a remarkably strong, yet explosively disintegrating piece of glass, the Prince Rupert's drop (also known as "Dutch tears")[25–27]. Unlike the studies mentioned above, we find that Prince Rupert's drops undergo a self-sustained fragmentation process driven by internal stresses only. We use high-resolution micro-computed X-ray tomography (micro-CT) to show that a mm-sized drop explodes into more than 20.000 pieces and measure its fragment size distribution down to 50 μm. Surprisingly, we find that the size distribution is not power-law, but has fragments that follow an exponential size distribution. Since an exponential size distribution naturally has a characteristic length as a fit parameter, this intrinsic length scale shows that the fragments, in this case, are of a well-defined size. We show that this parameter can be understood theoretically, and is directly linked to the residual stress within the Prince Rupert's drop. Comparing the outcomes with fragmentation measurements on other stressed and unstressed (glass) systems, allows us to show that two different types of breakup processes exist; random and hierarchic, each with their distinct size distribution. Our results provide a direct answer to the question of why power-law distributions are so ubiquitous, while at the same time they explain why size distributions are exponential in some other cases.

## Results and discussion

The Prince Rupert's drops are made by melting the end of a glass rod with a high thermal expansion coefficient and letting the resulting red hot droplets fall into a beaker with cold water. The rapid cooling causes thermal contraction, and the fact that the outside of the drop cools before the inside then leads to very large tensile stresses in the droplet center and compressive stresses on the exterior. The very large compressive forces suppress crack growth, giving rise to the drop's extreme strength: they do not break when hit with a hammer; in controlled experiments, Prince Rupert's drops can withstand loads of more than 10 kN without breaking[27]. If, however, a crack is initiated and it is able to reach the tensile central region, by e.g. breaking the tail of the droplet, it quickly propagates, causing the whole droplet to explosively disintegrate (Fig. 1a). The unique combination of their extreme strength and unstable nature has made Prince Rupert's drops the topic of scientific study since the first part of 17th-century[28]. In our experiments, to measure the fragment size distribution of a fragmented Prince Rupert's drop, we break a single drop inside a latex glove so that no secondary fractures can occur and all fragments can be easily collected. As a generic example, our micro-CT analysis (Fig. 2) shows that a millimeter-sized drop breaks up into 22,000 fragments. Figure 1d shows the distribution of the equivalent spherical diameters $d$, down to 50 μm, which is the maximum resolution of the micro-

CT. This fragment size distribution and all the others throughout the paper are represented by a probability density as a function of the equivalent spherical diameter. Though there are many ways of presenting fragment size data, this way allows one to directly compare length scales and enables a clear separation between data points when changes in the size distribution happen over a short interval such as this distribution for the Prince Rupert's drop (Fig. 1 d). The main conclusions are of course not changed by the way the data is presented.

The distribution of the Prince Rupert's drop (Fig. 1d), reveals two distinct exponential regimes, with two associated characteristic length scales ($d_1 = 0.31$ mm and $d_2 = 0.064$ mm), suggesting an inhomogeneous fragmentation. This result seems in conflict with a previous study of Prince Rupert's drops, where the fragment size distribution was fitted with a power law[29], but a direct comparison in fact shows the data is comparable (Supplementary Note 4). In that study, a sieving technique was applied. Our more precise micro-CT technique, allows each individual fragment to be accurately measured and leads to a fundamentally different conclusion. We measured each fragment size with great accuracy, i.e., all 20.000+ fragments were measured up to a size of 50 μm. In addition, Silverman et al. mixed fragments from different Prince Rupert's drops of a range of sizes, while we measure fragments from individual drops. The mixing of the fragments from different drops likely obscures the sharp transition between the two exponential regions that we find and broadens the size distribution. This together with more noisy data due to the limitations of the sieving technique, likely explains why Silverman et al. fit an power law to their data, while our data show that for an individual drop an exponential fit is more appropriate. The difference in glass type between the two studies is not likely to play any role, as will become clear in the following, our conclusions do not depend on the particular used material type, nor does it depend on the precise shape. 2D shapes of different glass types are found to break in a similar way to the more complicated 3D shape of the Prince Rupert's drop.

Analyzing the micro-CT images from a Prince Rupert's drop fractured inside a Carbopol matrix with the typical consistency of hairgel[30] so that it has sufficiently high yield stress to prevent the fragments from flying away (Supplementary Note 1), we find that the smallest fragments belonging to the first exponential regime are spread homogeneously throughout the broken droplet, indicating that these fragments are due to a secondary fragmentation process. Note that the small fragment regime is referred to in other studies as the "remaining dust". The micro-CT analysis makes it possible to determine the distribution of these fragments as well. However, the main conclusion is that the 97% of the mass of the drop contained in the second regime, exhibits an exponential fragment size distribution, which points to a random Poisson process, though theoretical efforts suggest that the process could also be a Matérn Hardcore Point Process due to the fact that fragments are not stress free[31]. As the difference is subtle, our work does not allow for this distinction. Interestingly, the Prince Rupert's data show resemblance with Mott's data of fragmented munition, where the distribution is also exponential, with the argument of the exponential equal to the square root of the fragment area[32]. Though in this research only one type of Prince Rupert's drop was used, an extended study could focus on including data of multiple drops, measuring stress levels before and after fragmentation as well as using different drop sizes with different levels of tempering. This can be achieved by using different glass types with different thermal expansion coefficients and by changing the flameworking of the glass. The micro-CT measurements of the Prince Rupert's drops fragments are however extremely costly and time-consuming so that perhaps a different technique should be used in such a study.

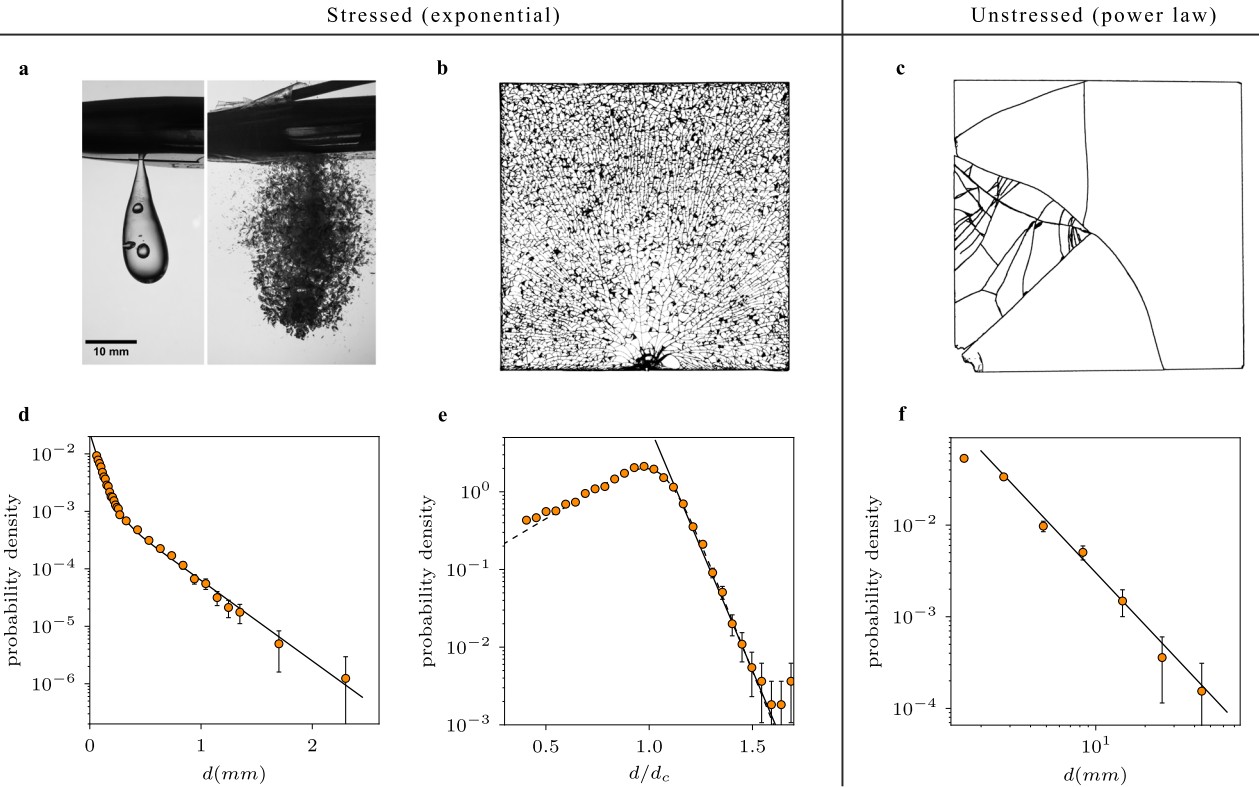

**Fig. 1 Fragment size distributions of stressed and unstressed materials.** The error bars in the graphs represent one standard deviation and are inferred by viewing the binning as a Bernoulli process. **a** Photograph of a Prince Rupert's drop clenched by a plier (left), and the fragmentation of a (different) Prince Rupert's drop triggered by cutting the tail with the plier (right). See Supplemental Information for a high-speed movie of the fragmentation of a droplet. **b** Tempered glass plate fractured by impact on one of its sides. For illustrative purposes the back of this particular plate is covered by adhesive tape so fragments are still fixed together after fragmentation, showing the crack pattern. **c** Fragmented unstressed glass plate, with the big fragments puzzled back together after fragmentation. The sizes that range from the size of the system all the way down to tiny fragments, illustrate well the power-law behavior in this case. **d** Fragment size distribution of the Prince Rupert's drop shown in **a**, where $d$ is the equivalent spherical diameter. One drop fragments in at least 21,847 fragments, measured by micro-CT. Two exponential regions can be identified, which indicates inhomogeneous fragmentation. The solid line is a fit of the form $p(d) \sim C_1 \exp(-d/d_1) + C_2 \exp(-d/d_2)$, with $d_1 = 0.31$ mm and $d_2 = 0.064$ mm. The smallest fragments are what in most other experiments is referred to as the "remaining dust". **e** Fragment size distribution of the tempered glass plates shown in **b** rescaled by $d_c$ for each plate thickness, where $d_c$ is the characteristic length set by the location of the maximum. The distribution is a truncated exponential distribution (as indicated by the solid line) with an exponential cutoff at a characteristic length set by $d_c$. The dotted line is a generalized logistic distribution, fitting the full distribution well. **f** Fragment size distribution of two unstressed glass plates, where the solid line is a power-law fit with an exponent of 1.9. Source data are provided as a Source Data file.

To understand the fracure data of the Prince Rupert's drop, we start by noting that Grady[33] showed that power-law distributions occur in brittle materials due to the excess build-up of elastic strain energies before the onset of fragmentation. When fragmentation finally occurs, cracks split at ever smaller length scales to dissipate the excess energy. Such a hierarchic breakup process is scale-free: goes on to smaller and smaller scales until all the energy is dissipated; this leads to the power-law size distributions that are observed in many materials. Though the power law suggests that no length scales underlie the fragmentation process, the boundaries of the power law can be indicative of such characteristic lengths still[34]. Our results however don't show any of such bounding length scales for which the power law holds and are therefore not considered. Furthermore, in other studies fragment size distributions sometimes show cross-over behavior between power-law and exponential regimes[35], also this is not observed in any of the measured size distributions. In case of the Prince Rupert's drop, the gently clipping off of the tail of the droplet does not lead to an excess build-up of elastic strain energy. In this case, a random breakup process occurs with crack branching happening at a well-defined length scale that is set by the level of internal stress, and the resulting size distribution is a simple exponential. This, therefore, calls for testing the hypothesis

of the existence of two distinct types of breakup processes: random (Poisson) and hierarchical.

To do so, we look at somewhat simpler, two-dimensional systems that are more amenable to theoretical analysis and numerical simulations: glass plates that are stressed, similar to the Prince Rupert's drop, and glass plates that are stress free or "unstressed", i.e., up to any small residual stress due to the manufacturing process that does not become apparent using crossed polarizers. In this case, one anticipates that the stressed glass plates to undergo a random breakup, just like the Prince Rupert's drops, while the unstressed plates are expected to follow a hierarchical breakup, and therefore both should differ strongly in their respective size distribution. For the stressed plates we use thermally tempered glass plates of six different thicknesses (3.83, 4.81, 5.88, 7.84, 9.87, 11.83 mm) and lateral size 24 cm × 24 cm, with the areas near the edge of the plates marked by paint in order to determine the origin of the fragments. The level of internal stress depends on the details of the tempering process, such as initial temperature and cooling rate. The complete stress profile is therefore determined and the tensile stress values are given in Methods. Fragmentation of the plates is initiated by initiating a single notch on the side of the glass, causing a fractured front to propagate and fragment the plate at an average

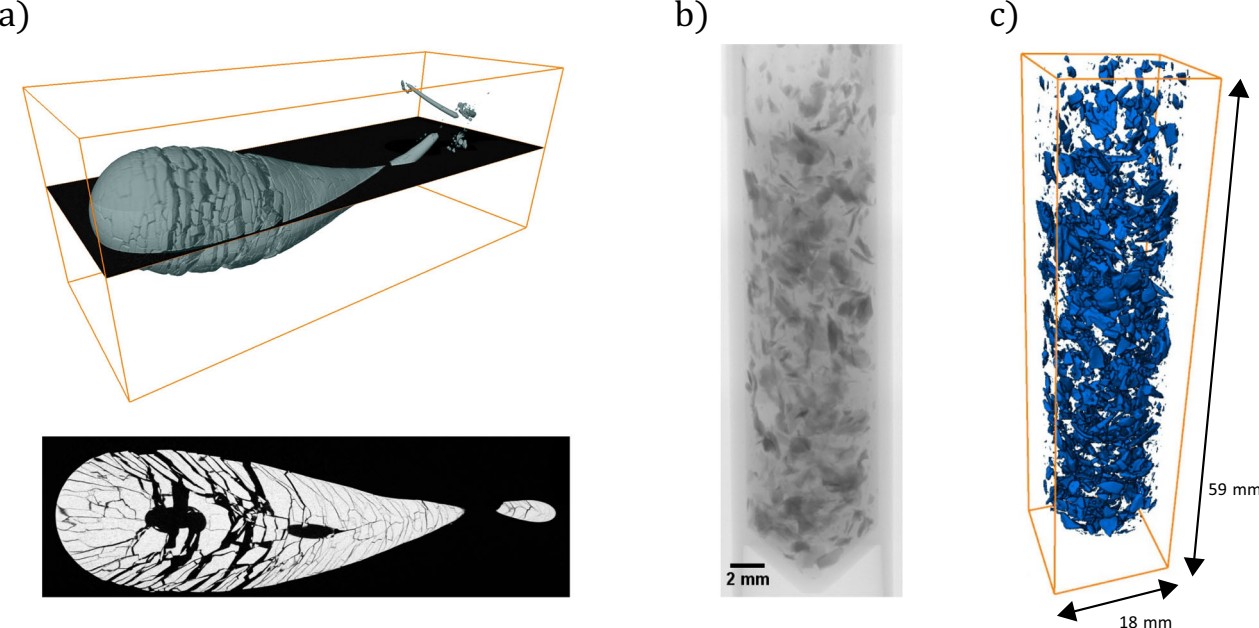

**Fig. 2 Prince Rupert's drops fragmented inside a Carbopol matrix. a** 3D reconstruction based on micro-computed tomography (micro-CT) data (top); cross-section as indicated by the black plane in the 3D image (bottom). There does not appear to be any clear spatial distribution of fragment sizes (see also Supplementary Information), except the arrangement of particles seems to have a structure similar to a pine cone. **b** X-ray image of some of the fragments inside the Carbopol matrix. **c** Example of a full 3D reconstruction used to determine individual fragment sizes.

speed of 1500 m s⁻¹ (see Fig. 1b and Supplementary Note 2), in agreement with previous observations[36]. For our analysis, we measure the fragment sizes of the plates with different thicknesses by weighing each piece individually. We find no significant difference in size between fragments from the paint-marked boundaries and those near the center of the plate. For each plate thickness, at least a thousand fragments were analyzed so that the total number of measured fragments is 11,626. Figure 1e shows the rescaled fragment size distributions. In agreement with our earlier hypothesis, we find an exponential distribution, truncated at a characteristic length scale $d_c$, which is approximately equal to the thickness of the plate. This is because fragments are restricted to a 2D plane and cannot be much smaller than the thickness of the plate. The truncated exponential can be very well fitted with a generalized logistic distribution (Fig. 1e) and coincides well with results from numerical simulations discussed below (for more details see Supplementary Note 3). The smallest "dust" particles produced in the plate experiments were not measured due to the limitations of measuring fragment sizes using a balance. For the unstressed plates, we used regular soda-lime glass of the same lateral size as the stressed plates and a thickness of 2.85 mm. The plates are fractured by throwing them on top of a thick glass plate at the bottom of a large plastic container, so all fragments could safely be collected. By throwing the plates with a sufficient velocity, enough kinetic energy was produced to ensure the formation of multiple fragments, something that a fall from a typical height (1–2 m) did not accomplish. It should be noted that this fragmentation method is different from how the breakup is initiated for the stressed glass plates. As the fragmentation of the tempered glass plates is entirely dominated by the internal stresses, the precise method of breakup is not important. Similarly, when the fragmentation is expected to be hierarchical, the precise method of fragmentation is also not important, as demonstrated by the ubiquitous presence of power-law distributions in different fragmentation experiments[24]. For creating a proper size distribution, however, it is necessary to have a sufficient amount of fragments, so that certain methods are

preferred over others. Figure 1c shows a broken plate where the biggest fragments are puzzled back together. The fracture pattern shows widely different sizes, exhibiting the typical power-law behavior one expects for 'normal' brittle fragmentation[18–23], where fragment sizes range from the order of the system, all the way down to indiscernibly small pieces. Figure 1f shows the size distribution of fragments from two of such plates, which is indeed power-law.

We now understand why for many fragmentation processes the breakup is generically hierarchical: there is so much energy supplied in breaking the material, that the system seeks for all possible ways of creating extra cracks to dissipate energy. This implies that there is no characteristic length scale and therefore no well-defined fragment size. If one wishes to either avoid dangerous large fragments (as is done for tempered safety glass) or otherwise to control the process and obtain a well-defined fragment size, one has to understand what gives the characteristic length scale of the exponential size distribution. To be able to predict this parameter for our stressed systems, and with that the average fragment size, we can use an energy-based argument that equates the elastic energy that was stored with the surface energy created due to fragmentation. Gulati[37] proposed a model for tempered glass plates where only the tensile part of the stored elastic energy is used for the creation of a new surface, though there are reasons to believe this assumption is not completely accurate[31,38]. The predicted characteristic fragment size $x$ for a plate of area $S$, if all the potential energy is used to create new fracture surface is then

$$x = \sqrt{\frac{S}{N}} = \frac{4}{\alpha}\left(\frac{\kappa_{Ic}}{\sigma_t}\right)^2, \qquad (1)$$

where $N$ the number of fragments per area $S$, $\sigma_t$ the tensile stress, $\kappa_{Ic}$ the mode-I fracture toughness and $\alpha = 16/(15\sqrt{3}(1+\nu))$ with $\nu$ the Poisson ratio. This presupposes that after fragmentation fragments contain no residual stress, which is unrealistic; using crossed polarizers to observe birefringence, we can see that

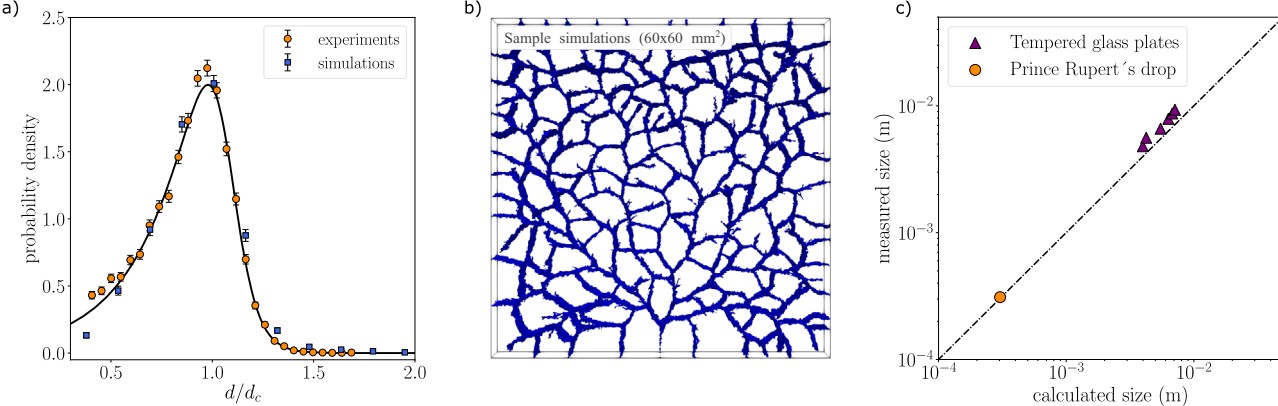

**Fig. 3 Predictions and measurements. a** Rescaled fragment size distributions of the tempered glass plates (orange circles) and data from finite-element simulations[39] (blue squares). Error bars indicate one SD. The truncated exponential can be fitted with a generalized logistic distribution (solid line). **b** Crack pattern of a simulated fracture in a tempered plate of 60 mm × 60 mm. The fragmented plate shows a strong resemblance with the experimental plate (Fig. 1b). **c** Measured fragment sizes against calculated fragment sizes. For the experimental fragment sizes the cross-over length of the exponential distributions are used. For the tempered glass plates these are the loci of the maximum $d_c$, while for the Prince Rupert's drop this is the onset of a secondary fragmentation process at $d = 310$ μm, i.e., the formation of "dust" particles. The predicted fragment sizes are calculated with Eq. (1) for the Prince Rupert's drop and Eq. (2) for the tempered glass plates. Source data are provided as a Source Data file.

the fragments still carry a fraction of the initial stress. This is a generic observation and results in an underestimation of the fragment size, simply because not all of the energy is used to create fragments. To correct this, it is common practice to use $\alpha$ as an adjustable parameter instead. We find that $\alpha = 0.19$, relatively close to the actual value for our glass type, i.e., $\alpha = 0.5$, yields accurate predictions of experimental fragment sizes. The value 0.19 is in fact a commonly found value (see[39] and references therein). To account also for the characteristic size in the Prince Rupert's drops we employ the above equation, using the tensile stress values measured by Aben et al.[27], as standard techniques do not allow to measure the stress values due to the drop's complicated shape. The measured values for the internal stresses of the Prince Rupert's drops are more than an order of magnitude larger than those in the glass plates, explaining why there are much more and much smaller fragments for the drops compared to the plates. Quantitatively, we find that although it is perhaps a bit naive to extend the plate model to a 3D fracture in this way, the prediction with the same fit parameter $\alpha = 0.19$, matches well with the characteristic size as found in our measurements (Fig. 3c). Note that the prediction for the Prince Rupert's drop (Eq. (1)) only applies to the large fragments of the distribution; the smaller "dust" particles also have an exponential size distribution, but with a characteristic size that is roughly an order of magnitude smaller. Our approach does not allow to account for these smaller fragments.

In order to not depend on the parameter $\alpha$, one can construct an improved model for the plates by assuming more realistically that the fragments are only stress-free up to a certain distance $\phi t$ from their edge, with $t$ the thickness of the plate[40]. According to this model, the fragment size is given by

$$x = \sqrt{\frac{S}{N}} = \frac{f\phi^2 t^2}{(f\phi t - 1)}, \tag{2}$$

where $f = \alpha(\sigma_t/\kappa_{Ic})^2$. Normally, the mean fragment size for an exponential distribution is set by the characteristic length scale $d$ of the exponent. However, for the glass plates, the distributions are truncated in the far tail of the curve, making $d$ an inappropriate length scale. Therefore, as the experimental fragment size, we use $d_c$, which sets the truncation point and maximum of the distribution. By using measured tensile stress values, we find that Eq. (2) yields good predictions for the fragment sizes of the

different glass plates with $\phi = 0.41$, the value proposed by Warren[40] (Fig. 3c). Following Eq. (2) we can make an order of magnitude estimate for the amount of fragments. A typical plate that has a tensile stress of 50 MPa and a thickness of 5 mm will according to this equation fragment in ~50,000 fragments per square meter, in line with our experimental observations, which indicate that fragment sizes are approximately equal to the plate thickness, which yields 40,000 fragments per square meter.

The fundamental difference in size distributions between the unstressed and stressed systems, the first being scale-free (hierarchical breakup) and the latter possessing an internal length scale (random breakup), confirms the existence of the two distinct breakup processes. To show that the power-law behavior, as seen in so many fragmentation events, is really due to an excess build-up of strain energy, we demonstrate the generality of our conclusions by breaking the same material in different ways and obtaining a power-law or exponential distribution by controlling the fragmentation process, hence controlling the size distribution in fragmentation. For these experiments, we use sugar glass in the form of disks with a diameter of 78 mm (for details on the preparation see Methods: Sugar glass disks for random and hierarchic breakup). The material is very similar (in breaking) to regular glass, but is easier to mold into a given shape due to its lower melting point. To accomplish a slow and controlled fragmentation due to internal stresses, we attach a sugar glass disk to a quartz glass plate of a much lower thermal expansion coefficient. By slowly cooling down the sample, from ~160 to 7 °C, the mismatch in thermal contraction will cause the sample to slowly crack, with more cracks appearing as the temperature is lowered. Figure 4a shows an example of the resulting crack pattern. The distribution of fragment sizes (Fig. 4c) is again exponential, confirming that the slow crack formation is a random (Poisson) process. The fast impact experiment on the same sugar glass disks again gives a power-law size distribution (Fig. 4b, d). This confirms our earlier hypothesis, and shows that the size distribution can be controlled. Moreover, the results show that the size distribution does in particular not depend on the type of material, but only the breakup mechanism (hierarchical or random) as demonstrated by the different breakup in the same material as well as the various other fragmented systems.

In sum, our results show that the self-sustained fragmentation of Prince Rupert's drops and tempered glass plates both give rise

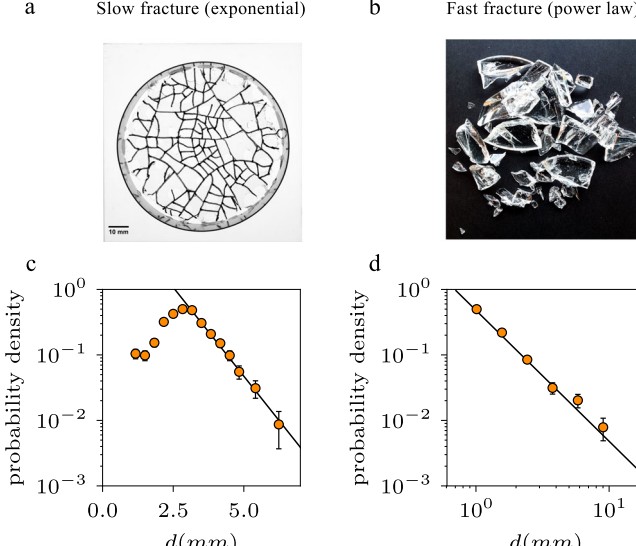

**Fig. 4 Slow versus fast fracture.** Fragmentation of sugar glass disks in a slow controlled way (left) compared with a fast catastrophic fragmentation (right). **a** A sugar glass disk, slowly fragmented using a mismatch in thermal expansion between the bottom plate and the sugar glass. The fracture lines are marked to make them clearly visible. **b** Fragments of a sugar glass disk fragmented by the impact on a solid surface. **c** The fragment size distribution of the slowly fractured sugar glass disks as the one shown in **a**, with error bars indicating the SD. The distribution is exponential with a characteristic length scale $d_1 = 0.8$ mm (as indicated by the solid line) and a cutoff at ~3 mm, confirming that this breakup is consistent with a random process. **d** Fragment size distribution of the dropped disks as one shown in **b**. This breakup, which represents the typical fragmentation study, is indeed power-law, with an exponent of 2. This shows that you can fragment the same material in different ways to have either random breakup (left) or hierarchical breakup (right). Source data are provided as a Source Data file.

to an exponential size distribution, and therefore a well-defined fragment size. This is surprisingly different from the results of most fragmentation studies, where power-law distributions are ubiquitous, and such a characteristic fragment size is missing. We demonstrate by also breaking other (glass) systems, that these two fundamentally different fragment size distributions can be explained by recognizing the two different breakup processes that occur in both cases. In normal brittle fragmentation, strain energies exceed the values that are needed for equilibrium fragmentation, so that when cracks appear, the breakup process is hierarchic, with crack branching at ever smaller length scales to dissipate the excess energy. This process is scale-free and results in a power-law distribution. For the Prince Rupert's drops that fracture due to internal stresses, the crack formation is in accordance with a random Poisson process, with the stress value determining the length scale at which crack branching occurs. It should be noted however that resulting fragments are not stress-free, hence the adjusted value of $\alpha$ in predicting fragment sizes, so that other theories such as the Matérn Hardcore process could not be ruled out. Interestingly, random breakup, such as for the Prince Rupert's drops, leads to well-defined fragment sizes. This may have strong repercussions for controlled fragmentation of objects, especially when a specific fragment size is desired.

## Methods
**Production of the Prince Rupert's drops**. The Prince Rupert's drops are made of a soda-lime type of glass (AR-GLAS) of the following composition: $SiO_2$, 69%; $B_2O_3$, 1%; $K_2O$, 3%; $Al_2O_3$, 4%; $Na_2O$, 13%; BaO, 2%; CaO, 5%; MgO, 3%. This

type of glass has a relatively high linear expansion coefficient ($9.1 \times 10^{-6}$ K$^{-1}$), which is needed to produce the large stresses in the quenching process. The droplets are formed by melting the end of a rotating solid glass rod with a torch until a "blob" of glass of sufficient size is formed. The rod is then held vertically for the drop to pinch off and fall into a beaker filled with room-temperature water. Frequently, the glass droplet does not survive the quenching process and breaks either quickly after impact or after a longer period of cooling (~30 s). The droplets that do survive are quite stable, and only break when the tail is cut off close to the main body of the droplet. This can be attributed to the fact that most of the tail is very thin and free of internal stresses.

**Stress measurements**. The stress in Prince Rupert's drops and glass plates is measured with a photoelastic scattered light polariscope (SCALP)[7,27]. The stress curves for thermally tempered glass plates are all parabola-shaped, with one parameter setting the value of the maximum stress. For other systems such as the Prince Rupert's drops as well as chemically tempered glass plates, the shape of the stress curves are not parabolic. This however does not seem to play any crucial role in the type of breakup process.

The measured tensile stresses of the plates for the different thicknesses are 54.3 ± 2.7 MPa (3.83 mm), 51.6 ± 2.6 MPa (4.81 mm), 45.6 ± 2.3 MPa (5.88 mm), 42.2 ± 2.1 MPa (7.84 mm), 41.5 ± 2.08 MPa (9.87 mm) and 42.7 ± 2.14 MPa (11.83 mm). For the Prince Rupert's drops, the stress curves have a similar shape, but are more irregular due to the more complicated shape of the droplet and the presence of vacuous bubbles. Compressive stresses can be as high as 700 MPa, thus explaining the enormous reinforcement that these glass droplets possess[27].

**Fragment size measurements**. To measure fragment sizes after fragmentation of a Prince Rupert's drop, a few of the biggest fragments were manually weighed. The remaining fragments were placed in a polymer solution (2 wt% Carbopol plus surfactant sodium dodecyl sulfate). After carefully mixing to prevent further fragmentation, the solution was gelled, mixed further, and divided over six plastic tubes. Using micro-CT, 3D reconstructions were made of the fragments (Fig. 2). The difference in density of the matrix (Carbopol plus air bubbles) and the glass particles is such that the images can be segmented efficiently using gray-level thresholding. The polymer gel matrix provides for an easy separation of individual fragments, something that is not possible by a simple stacking of fragments in the sample tubes. We determine the volume of each fragment, and calculate the fragment size using an equivalent spherical diameter $d$. The cumulative mass of the fragments was compared with the initial weight of the droplet before fragmenting, and the missing mass was on the order of 1%, which is well within experimental accuracy.

Since most of the fragments of the unstressed glass and sugar glass systems are large, fragments were measured by weighing. For the slowly fractured sugar glass, however, fragments remain attached to the bottom plate, so that fragment sizes had to be determined by other means. By marking the cracks and photographing the samples, the sizes of the fragments are determined through image analysis. Fragments on the border of the disk are excluded to rule out any possible edge effects. This procedure, resulting in about a 100 fragments per plate, is repeated several times until a total number of 1000 fragments sizes was obtained (Fig. 4a, c).

**Sugar glass disks for random and hierarchic breakup**. To perform experiments wherein the same brittle material, is broken in two different ways, we use sugar glass. This material closely resembles regular glass, yet it is safer and easier to process. We use Isomalt, a sugar alcohol frequently used as a sugar substitute. The material is heated to 167 °C to melt and can then be poured into molds of the shape of a disk. For the slow fracture of the glass disks, a layer of molten sugar glass is poured inside a metallic ring that is placed on top of a scratched quartz glass plate, which is heated by a hotplate of 165 °C. The scratches are needed to guarantee a good bond between the glass and the sugar glass. Once poured and spread equally over the mold, the sample is allowed to cool down to room temperature while it is kept inside a dry chamber with a relative humidity <2%. Next, the sample is placed inside a fridge for further cooling. Due to the large mismatch in thermal expansion coefficient between the sugar glass and the quartz glass, the cooling process causes the sugar glass to contract relative to the quartz glass and develop cracks one after the other (Fig. 1c).

For the fast fragmentation of sugar glass disks, the molten sugar is poured in metallic molds, cooled down, and kept inside the dry chamber. After one day, the samples are removed from their molds and are fragmented the same day to make sure that possible aging processes do not affect the results. All samples are only briefly removed from the dry chamber to allow for the fragmentation process, and quickly transferred back to the chamber, preventing the samples from absorbing any water. The disks are fragmented by dropping them on a metallic block from a 1 m height. The metallic block is surrounded by a plastic bag so that all fragments can easily be collected.

## Data availability
All data are available from the authors upon reasonable request. Source data are provided with this paper.

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

## Acknowledgements

This work is part of the Industrial Partnership Program Hybrid Soft Materials that is carried out under an agreement between Unilever Research and Development B.V. and the Netherlands Organisation for Scientific Research (NWO). We also want to thank GlassStress Ltd, in particular Mart Paemurru, for providing their expertise by measuring stress profiles of our tempered glass plates.

## Author contributions

S.K. and D.B. conceived the main concepts. G.v.D. performed the micro-CT imaging and the micro-CT analysis. S.K. did the experiments and analysis. J.-F.M. performed the numerical simulations on tempered glass plates. S.K. and D.B. wrote the manuscript.

## Competing interests

The authors declare no competing interests.
