## [Peer Review File · Nature Communications]

Reviewers' Comments:

Reviewer #1:

Remarks to the Author:

The authors offer for publication an interesting and thought-provoking study on the strain-energy induced fragmentation of brittle solids. The principal objective is to offer insights into, and explain, a dichotomy in the resulting fragment size distributions, and the statistical representation of the distributions, that has eluded the community for some number of decades. Namely, the hierarchical fragment distribution character observed in much experimental data, and the Poisson-like representation favored for other data. The experimental methods developed and applied by the authors to explore this issue are innovative, quite original and effectively address the issue of interest. Explanations offered by the authors are thought provoking. The manuscript warrants publication subject to some reflection and response to the comments and concerns in what follows.

History of the dichotomy immerses in the early parts of the last century. Applications in the geosciences, and in mining and concrete engineering, found that linear log-log plots of cumulative fragment mass versus size adequately described the particle size data being generated. Wartime efforts of Nevill Mott on munition fragmentation perhaps first popularized application of an exponential cumulative number versus square root mass (areal particle size) for the description of selected fragmentation data. Both representations were dictated by the experimental particle size assessment methods applied at the time.

The present authors choose to represent their statistical fragment size data differently. Since the earlier historical works provide the principal impetus for the present study, some explanations of the authors' representation of the fragment size data, and how it relates to the earlier work, are warranted.

In addition, the authors refer to their representation of the fragmentation particulate informally as a fragment size distribution. This is not inappropriate. However, nowhere in the principal text is the representation referred to as a fragment size density distribution. This distinction is left to the reader to discover on the vertical axes in the figures. Some up-front discussions of the authors' statistical representation of the data and their data analysis methodology would be appropriate.

The authors refer to fragment size data that plot linear log-log as hierarchical, or scale invariant. Moreover, that no length scale underlies the fragmentation physics. Other authors, perhaps more succinctly, have identified two length scales, or a schism in length scales, such that at sufficient energy intensities fragmentation becomes asymptotic scale invariant throughout the size span dictated by the bounding length scales. Some expansion by the authors on this issue would be informative.

The authors offer two explanations for the hierarchical versus Poisson dichotomy. One relates to that of contact-free internal strain energy fueling fragmentation, as opposed to impact or deformation imposed strain energy. Second, is that the strain energy density leading to hierarchical fragmentation is inherently more intense. Are both explanations necessary? Or, is it that contact-free internal strain fueling fragmentation is just inherently strain energy limited?

This reviewer would wish the authors to consider contrasting their study of the breakup of the Prince Rupert drop with that of Bergstrom et al (1961) fragmentation of glass balls. The works of Gilvarry are referred to (principally theoretical) but not the epic experimental efforts of Bergstrom and coworkers.

Bergstrom HC, Sollenberger CL, Mitchel W, (1961) Energy Aspects of Single Particle Crushing, Trans. AIME, 220, 367-372.

Does strain energy imparted to a glass sphere in that of Bergstrom differ fundamentally from that within the Prince Rupert drop? Does an internal fast-moving fracture know the difference? How

different in magnitude are the strain-energy densities fueling fracture and fragmentation in the two cases? If one prepared a Prince Rupert drop with an order of magnitude higher strain energy (not likely), and/or Bergstrom flawed the glass ball such that failure occurred at an order of magnitude lower strain energy, would the distributions start to be comparable? Or not? If not, why?

Again, this is an interesting and quite thought-provoking paper addressing the ongoing topic of solid-state fragmentation. I recommend publication, with the authors addressing to some level my concerns outline above.

Sincerely,

Dennis Grady

Reviewer #2:

Remarks to the Author:

The paper is interesting, worth publishing after revision and addresses a phenomena that is not yet fully understood in relation with the fracture of brittle solids. The conclusion that two different breakup processes exists is questionable from the data presented in the paper.

The authors could improve the paper as they did not consider background works from fractography of ceramics and glasses, glass tempering and recent research and related literature on the fragmentation process of glass, in particular thermally tempered glass similar to the fragmentation described here. Moreover the authors do not seem to be familiar with the research that has been done in the recent decades in the field of dynamic fracture mechanics. Therefore, the paper should be revised taking into consideration the following aspects:

page 3, line 57-59

The amount of residual stress from the sudden cooling of Prince Rupert's drop is expected to scatter significantly and also locally within a drop, especially due to different effects during the cooling that have an effect on the heat transfer to the surrounding medium, e.g. depending on the time of falling in the air, entering the water, the size and speed of the drop, turbulence at the interface between water, vapour, glass in the water, bubbles, etc., etc.. This cooling process is very different from the cooling of tempered glass by quenching with air as shown also. As the resulting residual stress is a governing parameter for the fragmentation size, did you measure the residual stress of each drop individually at different locations and perform a statistical analysis? If the stress was not measured it remains unclear how you want to calculate it back from fragmentation analysis which is a far more complex process. The assumption that the compression stress was 700 MPa from the paper [25] is not enough. In this context: How did you analyze the geometrical properties of each drop before fragmentation and what is the resolution and measurement error you can achieve for the geometrical properties? Please explain.

page 4, line 68

The assumption that this is a random Poisson process is not correct. It is much more likely a Matérn Hardcore Point Process (MHP) due to the fact that the amount of remaining residual stresses in each glass particle still existing after fragmentation prohibits a random Poisson process. Please see the following publications that you did not consider yet:

Nielsen, J. H. and Bjarrum, M. (2017): Deformations and strain energy in fragments of tempered glass: experimental and numerical investigation, *Glass Structures & Engineering* vol. 2.2, pp. 133–146.

Nielsen, J. H. (2017): Remaining stress-state and strain-energy in tempered glass fragments, *Glass Struc & Eng* 2, pp. 45–56.

Pourmoghaddam, N., Kraus, M.A., Schneider, J. et al. (2019) Relationship between strain energy and fracture pattern morphology of thermally tempered glass for the prediction of the 2D macro-scale fragmentation of glass. *Glass Struct Eng* 4, pp. 257–275.

Kraus, M.: Machine Learning Techniques for the Material Parameter Identification of Laminated Glass in the Intact and Post-Fracture State. PhD-Thesis. Universität der Bundeswehr München, 2019. <https://athene-forschung.unibw.de/doc/127852/127852.pdf>

Pourmoghaddam, N. (2020) On the Fracture Behaviour and the Fracture Pattern Morphology of Tempered Soda-Lime Glass. PhD-Thesis. TU Darmstadt. <https://www.springer.com/de/book/9783658282059>

page 4, line 82-83

Please add the amount of surface compression stress measured for each plate and their statistical distribution (if different points were measured). As it is a very good model in the publications mentioned above that the fragmentation depends on the strain energy density and thus on the square of the stress level and linearly on the thickness, this information is very important. Moreover, as the stress level that can be technically induced by the tempering process very much depends on the heat transfer coefficient generated by the blowers used for quenching in the tempering machine and the related air pressure, the term "thermally tempered" can be misleading in terms of the amount of the residual stress really present after the tempering process, which varies from heat strengthened to tempered glass, see also your variation of the stresses (page 20, line 337). Please have a look in the following papers:

Narayanaswamy, O. S. (1978): Stress and structural relaxation in tempering glass, in: *Journal of the American Ceramic Society* vol. 61.3-4, pp. 146–152.

Narayanaswamy, O. S. (2001): Evolution of glass tempering models, in: *proceedings Glass Processing Days, International conference on architectural and automotive glass, Tampere, Finland*, pp. 83-86.

Carré, H. and Daudeville, L. (1996): Numerical Simulation of Soda-Lime Silicate Glass Tempering, in: *Le Journal de Physique IV* vol. 6.C1, pp. 175–185.

Nielsen, J. H. (2009): *Tempered Glass: Bolted Connections and Related Problems*, PhD thesis, Technical University of Denmark. <https://backend.orbit.dtu.dk/ws/portalfiles/portal/5433989/Tempered+Glass.pdf>

Pourmoghaddam, N. and Schneider, J. (2019): Determination of the engine power for quenching of glass by forced convection: simplified model and experimental validation of residual stress levels, in: *Glass Struct Eng* 4, pp. 117–125.

page 4, line 86

This information is not new. Please compare your results to the following work which is also missing in the references:

Nielsen, J.H., Olesen, J.F. & Stang, H. The Fracture Process of Tempered Soda-Lime-Silica Glass. *Exp Mech* 49, 855 (2009)

More information on the fracture process and fractography, e.g. of the morphology of the glass surfaces after fracture can be found in:

Quinn, G. D. (2016): Fractography of Ceramics and Glasses, vol. 191, National Institute of Standards and Technology: <https://nvlpubs.nist.gov/nistpubs/specialpublications/NIST.SP.960-16e2.pdf>

page 5, line 110 - 112

The assumption that only the tensile part of the stored elastic energy is used for creation of new surface energy is questionable and thus also equation (1). See the following older and a recent publication on the relationship between the temper stress, the strain energy density and the fragmentation size:

Akeyoshi, K. and Kanai, E. (1965): Mechanical Properties of Tempered Glass, in: VII int. Congr. of Glass paper 80.

Pourmoghaddam, N., Kraus, M.A., Schneider, J. et al. (2019) Relationship between strain energy and fracture pattern morphology of thermally tempered glass for the prediction of the 2D macro-scale fragmentation of glass. *Glass Struct Eng* 4, pp. 257–275.

Page 6, line 135-139

The assumption that a typical tempered glass plate has a tensile stress of 50 MPa on average is true, but the sensitivity of the fragmentation size on the stress (as it depends to the square on the stress level and linearly on the thickness based on the assumption of the dependence on the strain energy density) does not allow the extrapolation that simple.

Page 6, line 141-143

I do not think that you can draw this conclusion from the experiments shown in this paper without knowing (a) what was the exact of strain energy in the particles and (b) how much of the initial strain energy was really converted to new surfaces depending on (c) the initial geometry of the drop and the stress distribution within the drop.

What is the total of the fracture surface that was generated after the fragmentation of a drop? Can you make an assumption on that? Please also refer to the following works:

Fineberg, J. and Marder, M. (1991): Instability in dynamic fracture, in: *Physical Review Letters* vol. 67.4, pp. 457–460

Fineberg, J (2006): The Dynamics of Rapidly Moving Tensile Cracks In Brittle Amorphous Material, in: *Dynamic Fracture Mechanics*, ed. by A. Shukla, pp. 104–146.

Dugnani, R., Zednik, R. J., and Verghese, P. (2014): Analytical model of dynamic crack evolution in tempered and strengthened glass plates, in: *International Journal of Fracture* vol. 190.1-2, pp. 75–86

page 7, line 157 - 168 Conclusions

The conclusions are not correct seen before the aspects on (a) the remaining stress state in the particles and the uncertainty with regard to (b) the initial stress state of the drops and their distribution before fragmentation, (c) their geometrical size - see above. In particular, the assumption of a random Poisson process is not correct, it is more likely a Matérn hardcore point process (MHP), see also:

Dereudre, D. and Lavancier, F. (2011): Practical simulation and estimation for Gibbs Delaunay-Voronoi tessellations with geometric hardcore interaction, in: *Computational Statistics and Data Analysis* vol. 55.1, pp. 498–519.

Martinez, W. L. ; Martinez, A. R.: *Computational Statistics Handbook with MATLAB*,

Third Edition -. Boca Raton, Fla: CRC Press, 2015. – ISBN 978–1–466–59274–2

Pourmoghaddam, N., Kraus, M.A., Schneider, J. et al. (2019) Relationship between strain energy and fracture pattern morphology of thermally tempered glass for the prediction of the 2D macro-scale fragmentation of glass. *Glass Struct Eng* 4, pp. 257–275.

page 10, line 249-254

Please explain more in detail if and how the Scalp instrument was used to measure the stress for the drops that were the subject of your investigation. Typically, the size of the prism of the Scalp instrument is too large to make it possible to make a measurement on a small drop. Also, please explain the scattering of the results of each individual measurement done at different locations for each drop and also for the tempered glass (see above).

page 13, figure 1, also page 11, line 266

It is not correct that the "unstressed" glass plate did not have stress. First of all, in annealed glass, depending on the glass thickness typically residual stresses from the float process in the range of 3-10 MPa still exist. Secondly, the fracture initiation itself induces very large local contact stresses and an elastic wave running through the glass plate and getting reflected at the edges. This influences the fracture pattern as also can be seen from your figures 3 (b), (c) and 4 (b). Please add the information how much force was used to break the glass and what was the material to break it, its properties and what was the contact radius of the breaking tool.

page 14, figure 2

Also here, it would be interesting to understand better (a) how much stress is induced by the "slow controlled way" and (b) how much residual stress existed before fragmentation.

page 16, figure 4

Please add the measured residual stress in the tempered glass plates shown as triangles. Again, equation (1) can be misleading if the remaining stress in the particles is not considered at all and if different stress levels exist before fragmentation. In Figure 4(c), how did you measure and calculate the (average) size?

page 18, line 314-323 and figure 6

Please compare your results to the following work:

Nielsen, J.H., Olesen, J.F. & Stang, H. The Fracture Process of Tempered Soda-Lime-Silica Glass. *Exp Mech* 49, 855 (2009)

page 19, line 324-page 20, line 347

The description of the Finite Element Analysis should be given in more detail and also illustrated as it is hard to understand from a pure verbal way.

Based on which information did you vary the residual tension stress between 35 MPa and 60 MPa? What is the underlying statistical distribution for the residual stress level and how did you compare it to your photoelastic measurements? Which thickness had which stress level? As the sensitivity of the fragmentation on the stress level is very high (see papers and research given above), a small difference in the stress level will have a very large effect on the fragmentation size. E.g., if you use a 3 mm thick glass and it is subjected to only 35 MPa it will result in a very coarse fracture pattern (strain energy depends linearly on the thickness) but with increasing stress it will change very strongly (strain energy depends to the square on the stress).

Modeling: How and where did you locate the cohesive zone elements and what was the element's geometry and size? As you have to put the cohesive zone elements in advance, the meshing will influence the fracture pattern ("self fulfilling prophecy"). And the element's size will have a minimum feasible size due to computation time. So the finite element calculation's boundary conditions will influence the results. It is not clear to what extent thus the boundary conditions you put in the finite element code lead to your interpretation of having different breakup processes

whereas in reality they are fundamentally the same but only on a different scale. Please discuss and explain better your FE model.

Reviewer #3:

None

Reviewer #4:

Remarks to the Author:

I have read the manuscript at least two times and in my opinion the manuscript is not in a state which can be recommended for publication in Nature Communications.

The most significant reason why I make this recommendation is that the authors have not clearly described their motivation for carrying out the work. If they wanted to repeat the findings of the work by Silverman et al. (2012), then they should have stated so.

The authors should have noted that in the studies carried out by Silverman et al. (2012), Prince Rupert's drops were made of a lead-crystal glass and they studied fragments collected from disintegration of ~ 50 drops of head diameters in the range of 10 to 15 mm and overall lengths of ~ 70 to 120 mm.

Although the authors investigated about 2 or 3 Rupert's drops of soda-lime glass of a head diameter of ~ 2.5 mm, a considerable part of their studies was made on the fragments generated by the disintegration of thermally tempered soda-lime glass plates of thicknesses in the range of 4 to 12 mm.

The authors believe that the fragmentation behaviour of a 2-d plate can be likened to the fragmentation behavior of a 3-D Rupert's drop. This is not correct since when a Rupert's drop explodes, many fragments fly out radially, whereas in the case of the fragmentation of a thermally tempered glass plate, the fragments do not fly away, but stay in their pre-disintegration positions (see, for example, Nature 320 (6057) 48-50 (1986)).

Another difference between the fragmentation behaviour of a Rupert's drop made of soda-lime glass and a Rupert's drop made of a lead-crystal glass is that the speed of propagation of disintegration in drops of soda-lime glass is significantly higher than that in Rupert's drops made of a lead-crystal glass. Such a difference may affect the fragment size distribution.

My suggestion to the authors is to conduct fragmentations of a much larger number of Rupert's drops made of soda-lime glass and a lead-crystal glass (not plates) and then analyze the results carefully before drawing their conclusions.

REVIEWER COMMENTS

Reviewer #1 (Remarks to the Author):

The authors offer for publication an interesting and thought-provoking study on the strain-energy induced fragmentation of brittle solids. The principal objective is to offer insights into, and explain, a dichotomy in the resulting fragment size distributions, and the statistical representation of the distributions, that has eluded the community for some number of decades. Namely, the hierarchical fragment distribution character observed in much experimental data, and the Poisson-like representation favored for other data. The experimental methods developed and applied by the authors to explore this issue are innovative, quite original and effectively address the issue of interest. Explanations offered by the authors are thought provoking. The manuscript warrants publication subject to some reflection and response to the comments and concerns in what follows.

Reply: Thank you very much for your very positive and constructive comments. We will reply to each of them below, and have changed the manuscript accordingly. We now thank you and the other referees in the acknowledgements for your very helpful contributions and comments.

History of the dichotomy immerses in the early parts of the last century. Applications in the geosciences, and in mining and concrete engineering, found that linear log-log plots of cumulative fragment mass versus size adequately described the particle size data being generated. Wartime efforts of Nevill Mott on munition fragmentation perhaps first popularized application of an exponential cumulative number versus square root mass (areal particle size) for the description of selected fragmentation data. Both representations were dictated by the experimental particle size assessment methods applied at the time.

The present authors choose to represent their statistical fragment size data differently. Since the earlier historical works provide the principal impetus for the present study, some explanations of the authors' representation of the fragment size data, and how it relates to the earlier work, are warranted.

In addition, the authors refer to their representation of the fragmentation particulate informally as a fragment size distribution. This is not inappropriate. However, nowhere in the principal text is the representation referred to as a fragment size density distribution. This distinction is left to the reader to discover on the vertical axes in the figures. Some up-front discussions of the authors' statistical representation of the data and their data analysis methodology would be appropriate.

Reply: we thank you for these interesting comments. We underline in the main text now that we represent the data as a fragment size density distribution, and explain in detail how we get this representation. We also discuss the Mott data in somewhat more detail in the revised version; we think that looking at the equivalent spherical diameter is better than looking at the area distribution, especially for the very heterogeneous shapes that we find. For instance in the distribution of the Prince Rupert's drop, there is

a short and sharp change in slope in the exponential at small length scales. Using the equivalent spherical diameter makes this transition clearly visible, while using another measure like the mass would probably obscure this.

The authors refer to fragment size data that plot linear log-log as hierarchical, or scale invariant. Moreover, that no length scale underlies the fragmentation physics. Other authors, perhaps more succinctly, have identified two length scales, or a schism in length scales, such that at sufficient energy intensities fragmentation becomes asymptotic scale invariant throughout the size span dictated by the bounding length scales. Some expansion by the authors on this issue would be informative.

We apologize, but have no good idea what literature the referee is referring to. Our paper does indeed present experimental evidence for the statement that a fragmentation at relatively low energies has with one or two length scales (the 'fine dust' of the prince Rupert's drops could be the second length scale). In the non-stressed systems the excess energy does then lead to a scale-free fragment distribution. Is this what you mean? Should we include an extra reference?

The authors offer two explanations for the hierarchical versus Poisson dichotomy. One relates to that of contact-free internal strain energy fueling fragmentation, as opposed to impact or deformation imposed strain energy. Second, is that the strain energy density leading to hierarchical fragmentation is inherently more intense. Are both explanations necessary? Or, is it that contact-free internal strain fueling fragmentation is just inherently strain energy limited?

We think that indeed both are necessary. To understand the difference, it is not only necessary to consider that indeed, as the referee suggests, internal strain fuels fragmentation for the pre-stressed systems, but also that there is a large excess strain energy for the hierarchical fragmentation. In essence this is of course the same argument: in the pre-stressed system there is just enough energy to create the fragments, in the non-stressed ones there is a large excess. This is part of the discussion in in the manuscript.

This reviewer would wish the authors to consider contrasting their study of the breakup of the Prince Rupert drop with that of Bergstrom et al (1961) fragmentation of glass balls. The works of Gilvarry are referred to (principally theoretical) but not the epic experimental efforts of Bergstrom and coworkers.

Bergstrom HC, Sollenberger CL, Mitchel W, (1961) Energy Aspects of Single Particle Crushing, Trans. AIME, 220, 367-372.

Due to the current corona-crisis our library services are not what they should be. We couldn't find this particular reference online. If this is considered necessary we would like to request the reviewer to send a copy to Nat Comms.

Does strain energy imparted to a glass sphere in that of Bergstrom differ fundamentally from that within the Prince Rupert drop? Does an internal fast-moving fracture know the difference? How different in magnitude are the strain-energy densities fueling fracture and fragmentation in the two cases? If one prepared a Prince Rupert drop with

an order of magnitude higher strain energy (not likely), and/or Bergstrom flawed the glass ball such that failure occurred at an order of magnitude lower strain energy, would the distributions start to be comparable? Or not? If not, why?

Again, this is an interesting and quite thought-provoking paper addressing the ongoing topic of solid-state fragmentation. I recommend publication, with the authors addressing to some level my concerns outline above.

Sincerely,

Dennis Grady

Reviewer #2 (Remarks to the Author):

The paper is interesting, worth publishing after revision and addresses a phenomena that is not yet fully understood in relation with the fracture of brittle solids. The conclusion that two different breakup processes exists is questionable from the data presented in the paper.

The authors could improve the paper as they did not consider background works from fractography of ceramics and glasses, glass tempering and recent research and related literature on the fragmentation process of glass, in particular thermally tempered glass similar to the fragmentation described here. Moreover the authors do not seem to be familiar with the research that has been done in the recent decades in the field of dynamic fracture mechanics. Therefore, the paper should be revised taking into consideration the following aspects:

We are very grateful for your positive evaluation, and all of your very interesting remarks. We have done our very best to answer these below, and have changed the manuscript accordingly. We now thank you and the other referees in the acknowledgements for your very helpful contributions and comments.

page 3, line 57-59

The amount of residual stress from the sudden cooling of Prince Rupert's drop is expected to scatter significantly and also locally within a drop, especially due to different effects during the cooling that have an effect on the heat transfer to the surrounding medium, e.g. depending on the time of falling in the air, entering the water, the size and speed of the drop, turbulence at the interface between water, vapour, glass in the water, bubbles, etc., etc.. This cooling process is very different from the cooling of tempered glass by quenching with air as shown also. As the resulting residual stress is a governing parameter for the fragmentation size, did you measure the residual stress of each drop individually at different locations and perform a statistical analysis? If the stress was not measured it remains unclear how you want to calculate it back from fragmentation analysis which is a far more complex process. The assumption that the compression stress was 700 MPa from the paper [25] is not enough. In this context: How did you analyze the geometrical

properties of each drop before fragmentation and what is the resolution and measurement error you can achieve for the geometrical properties? Please explain.

We have insisted heavily to try and measure the stresses using the same method as for the glass plates; however this turned out to be impossible. The estimate from the literature is thus our best guess. However this is exactly the reason why we looked into the tempered glass plates. Since there we could measure the stresses and compare to Eq.s 1 and 2, this allowed us to check whether we could also predict the number of fragments for the prince Rupert's drop. As fig.4c shows, this prediction is very reasonable, giving us confidence that we really nailed the problem.

page 4, line 68

The assumption that this is a random Poisson process is not correct. It is much more likely a Matérn Hardcore Point Process (MHP) due to the fact that the amount of remaining residual stresses in each glass particle still existing after fragmentation prohibits a random Poisson process. Please see the following publications that you did not consider yet:

Nielsen, J. H. and Bjarrum, M. (2017): Deformations and strain energy in fragments of tempered glass: experimental and numerical investigation, *Glass Structures & Engineering* vol. 2.2, pp. 133–146.

Nielsen, J. H. (2017): Remaining stress-state and strain-energy in tempered glass fragments, *Glass Struc & Eng* 2, pp. 45–56.

Pourmoghaddam, N., Kraus, M.A., Schneider, J. et al. (2019) Relationship between strain energy and fracture pattern morphology of thermally tempered glass for the prediction of the 2D macro-scale fragmentation of glass. *Glass Struct Eng* 4, pp. 257–275.

Kraus, M.: Machine Learning Techniques for the Material Parameter Identification of Laminated Glass in the Intact and Post-Fracture State. PhD-Thesis. Universität der Bundeswehr München, 2019. <https://athene-forschung.unibw.de/doc/127852/127852.pdf>

Pourmoghaddam, N. (2020) On the Fracture Behaviour and the Fracture Pattern Morphology of Tempered Soda-Lime Glass. PhD-Thesis. TU Darmstadt. <https://www.springer.com/de/book/9783658282059>

We thank the referee for bringing up this interesting point. We think a discussion of the Matern Hardcore Point Process is a bit too technical for the general audience that *Nature Comms.* is aiming for, but we now explicitly mention it, and state that the process we observe is perhaps not a perfectly random Poisson process because of the residual stresses, and have added the references so that the interested reader can follow up.

page 4, line 82-83

Please add the amount of surface compression stress measured for each plate and their

statistical distribution (if different points were measured). As it is a very good model in the publications mentioned above that the fragmentation depends on the strain energy density and thus on the square of the stress level and linearly on the thickness, this information is very important. Moreover, as the stress level that can be technically induced by the tempering process very much depends on the heat transfer coefficient generated by the blowers used for quenching in the tempering machine and the related air pressure, the term "thermally tempered" can be misleading in terms of the amount of the residual stress really present after the tempering process, which varies from heat strengthened to tempered glass, see also your variation of the stresses (page 20, line 337). Please have a look in the following papers:

Narayanaswamy, O. S. (1978): Stress and structural relaxation in tempering glass, in: Journal of the American Ceramic Society vol. 61.3-4, pp. 146–152.

Narayanaswamy, O. S. (2001): Evolution of glass tempering models, in: proceedings Glass Processing Days, International conference on architectural and automotive glass, Tampere, Finland, pp. 83-86.

Carré, H. and Daudeville, L. (1996): Numerical Simulation of Soda-Lime Silicate Glass Tempering, in: Le Journal de Physique IV vol. 6.C1, pp. 175–185.

Nielsen, J. H. (2009): Tempered Glass: Bolted Connections and Related Problems, PhD thesis, Technical University of Denmark.
<https://backend.orbit.dtu.dk/ws/portalfiles/portal/5433989/Tempered+Glass.pdf>

Pourmoghaddam, N. and Schneider, J. (2019): Determination of the engine power for quenching of glass by forced convection: simplified model and experimental validation of residual stress levels, in: Glass Struct Eng 4, pp. 117–125.

We agree, have added the data and their uncertainty. We also added a small discussion on the term 'thermally tempered, along the lines suggested by the referee, included the the references; we now explicitly state that the residual stress of course depends on the (details of the) tempering process.

page 4, line 86

This information is not new. Please compare your results to the following work which is also missing in the references:

Nielsen, J.H., Olesen, J.F. & Stang, H. The Fracture Process of Tempered Soda-Lime-Silica Glass. Exp Mech 49, 855 (2009)

More information on the fracture process and fractography, e.g. of the morphology of the glass surfaces after fracture can be found in:

Quinn, G. D. (2016): Fractography of Ceramics and Glasses, vol. 191, National Institute of Standards and Technology:
<https://nvlpubs.nist.gov/nistpubs/specialpublications/NIST.SP.960-16e2.pdf>

We fully agree, and had in fact added references to this extent in the discussion of the rapid fracture results (Formerly ref 33 and 34). We have added the new references, and yet another reference suggested by the other referee, and now also mention all these references on page 4.

page 5, line 110 - 112

The assumption that only the tensile part of the stored elastic energy is used for creation of new surface energy is questionable and thus also equation (1). See the following older and a recent publication on the relationship between the temper stress, the strain energy density and the fragmentation size:

Akeyoshi, K. and Kanai, E. (1965): Mechanical Properties of Tempered Glass, in: VII int. Congr. of Glass paper 80.

Pourmoghaddam, N., Kraus, M.A., Schneider, J. et al. (2019) Relationship between strain energy and fracture pattern morphology of thermally tempered glass for the prediction of the 2D macro-scale fragmentation of glass. *Glass Struct Eng* 4, pp. 257–275.

Thank you; we have added to the discussion that one should use Eq.1 with caution, especially for tempered systems, and have added the references.

Page 6, line 135-139

The assumption that a typical tempered glass plate has a tensile stress of 50 MPa on average is true, but the sensitivity of the fragmentation size on the stress (as it depends to the square on the stress level and linearly on the thickness based on the assumption of the dependence on the strain energy density) does not allow the extrapolation that simple.

This is nothing else than an order of magnitude estimate, which only goes to show that such a reasoning may lead one to relatively easily understand for instance the order of magnitude of the number of fragments of the broken glass of a bus stop. We thought this was insightful, and would allow readers to understand the basic arguments used in our paper. We have added to the revised version this is merely on order of magnitude estimation. If the referee thinks this is confusing, we can take it out.

Page 6, line 141-143

I do not think that you can draw this conclusion from the experiments shown in this paper without knowing (a) what was the exact of strain energy in the particles and (b) how much of the initial strain energy was really converted to new surfaces depending on (c) the initial geometry of the drop and the stress distribution within the drop. What is the total of the fracture surface that was generated after the fragmentation of a drop? Can you make an assumption on that? Please also refer to the following works:

Fineberg, J. and Marder, M. (1991): Instability in dynamic fracture, in: *Physical Review Letters* vol. 67.4, pp. 457–460

Fineberg, J (2006): The Dynamics of Rapidly Moving Tensile Cracks In Brittle Amorphous Material, in: *Dynamic Fracture Mechanics*, ed. by A. Shukla, pp. 104–146.

Dugnani, R., Zednik, R. J., and Verghese, P. (2014): Analytical model of dynamic crack evolution in tempered and strengthened glass plates, in: International Journal of Fracture vol. 190.1-2, pp. 75–86

We thank you for these comments, and have added the suggested references. We would like to underline that that we use the glass plates and sugar glass disks not only to show the generality of our results, but also to be able to have precise measurements, in the glass plates, for the initial and residual stresses. Which allows us to do the energy balance. This is at the basis of Fig.4c where we show that good agreement between the results of the glass plates and those of the prince Rupert's drops. This shows that the Eqs. 1 and 2 give a good prediction for the fragment sizes, which in turn gives us confidence in the results presented in our manuscript.

page 7, line 157 - 168 Conclusions

The conclusions are not correct seen before the aspects on (a) the remaining stress state in the particles and the uncertainty with regard to (b) the initial stress state of the drops and their distribution before fragmentation, (c) their geometrical size - see above. In particular, the assumption of a random Poisson process is not correct, it is more likely a Matérn hardcore point process (MHP), see also:

Dereudre, D. and Lavancier, F. (2011): Practical simulation and estimation for Gibbs Delaunay-Voronoi tessellations with geometric hardcore interaction, in: Computational Statistics and Data Analysis vol. 55.1, pp. 498–519.

Martinez, W. L. ; Martinez, A. R.: Computational Statistics Handbook with MATLAB, Third Edition -. Boca Raton, Fla: CRC Press, 2015. – ISBN 978-1-466-59274-2

Pourmoghaddam, N., Kraus, M.A., Schneider, J. et al. (2019) Relationship between strain energy and fracture pattern morphology of thermally tempered glass for the prediction of the 2D macro-scale fragmentation of glass. Glass Struct Eng 4, pp. 257–275.

We have added a discussion on the Poisson vs. Matern hardcore point process. We also stress again that we use the glass plates and sugar glass disks not only to show the generality of our results, but also to be able to have precise measurements, in the glass plates, for the initial and residual stress, as well as their geometrical size. The observed good agreement between the results of the glass plates and those of the prince Rupert's drops gives us some confidence that the results presented in our manuscript can be trusted also in this respect.

page 10, line 249-254

Please explain more in detail if and how the Scalp instrument was used to measure the stress for the drops that were the subject of your investigation. Typically, the size of the prism of the Scalp instrument is too large to make it possible to make a measurement on a small drop. Also, please explain the scattering of the results of each individual measurement done at different locations for each drop and also for the tempered glass (see above).

The Scalp instrument was only used to measure the stress in the glass plates for exactly this reason. For the Prince Rupert's drops we therefore use previous literature estimates. We underline this once more in the revised version of the manuscript.

page 13, figure 1, also page 11, line 266

It is not correct that the "unstressed" glass plate did not have stress. First of all, in annealed glass, depending on the glass thickness typically residual stresses from the float process in the range of 3-10 MPa still exist. Secondly, the fracture initiation itself induces very large local contact stresses and an elastic wave running through the glass plate and getting reflected at the edges. This influences the fracture pattern as also can be seen from your figures 3 (b), (c) and 4 (b). Please add the information how much force was used to break the glass and what was the material to break it, its properties and what was the contact radius of the breaking tool.

Thank you again; we now specify that the stress in the "normal" glass was so small that it did not show up between crossed polarizers; of course there is a small amount of residual stress from the manufacturing and we now acknowledge that. In the previous version, we did specify how we break the normal glass: "The plates are fractured by throwing them on top of a hard surface at the bottom of a large plastic container, so all fragments could safely be collected." In the revised version of the manuscript we give more details on this.

page 14, figure 2

Also here, it would be interesting to understand better (a) how much stress is induced by the "slow controlled way" and (b) how much residual stress existed before fragmentation.

Thank you; we have explained this better in the new version of the manuscript. The stress is given by the thermal mismatch between the quartz and sugar glass, this is controlled by controlling the cooling rate; the quartz has almost no thermal expansion, and so it is mainly the shrinking of the sugar glass disk that governs the stress. This can easily be estimated; the sugar glass cools down roughly 160 degrees after having been molded. The thermal expansion coefficient of the sugar glass is $\sim 10^{-4}/K$, meaning that the disk shrinks by approximately 2%. For an elastic modulus of 100MPa, this yields a stress of ~ 2 MPa. The breaking strain of sugar glass is a roughly a percent, so the residual stress is ~ 1 MPa.

page 16, figure 4

Please add the measured residual stress in the tempered glass plates shown as triangles. Again, equation (1) can be misleading if the remaining stress in the particles is not considered at all and if different stress levels exist before fragmentation. In Figure 4(c), how did you measure and calculate the (average) size?

Although we tried to measure the residual stress in the tempered glass plates fragments, we were not able to (because of their size). The residual stress is however considered for the tempered glass plates, it is part of the model. The model assumes that the resulting fragments are stress free up to a certain distance from the boundary. We used the value proposed by Warren and find that using this value yields good predictions of the fragment sizes. So the model takes the initial size and level of internal stress of the

plates that we measured, plus the parameter setting the residual stress given by Warren. For the fragment size we take the maximum in the size distribution in Fig. 4 (a).

page 18, line 314-323 and figure 6

Please compare your results to the following work:

Nielsen, J.H., Olesen, J.F. & Stang, H. The Fracture Process of Tempered Soda-Lime-Silica Glass. *Exp Mech* 49, 855 (2009)

Thank you for bringing this to our attention; we have added the reference. The results are in perfect agreement (front propagation velocity of ~ 1500 m/s) with our and earlier results that in the previous version were Refs. 33 and 34.

page 19, line 324-page 20, line 347

The description of the Finite Element Analysis should be given in more detail and also illustrated as it is hard to understand from a pure verbal way.

Based on which information did you vary the residual tension stress between 35 MPa and 60 MPa?

We have varied the residual tension stress between 35 MPa and 60 MPa because these are classically observed in tempered glass and were the values reported in the experiments of Mognato et al. [1] and Akeyoshi and Kanai [2]. They were also used in the analytical model of Warren [2]. The numerical fragmentation results were compared to these in our earlier work [31]. References [1-3] have been added to supplemental material.

[1] Mognato E, Barbieri A, Schiavonato M, Pace M. Thermally toughened safety glass: correlation between flexural strength, fragmentation and surface compressive stress. In: *Proceedings of glass performance days, Tampere, Finland; 2011*. p. 115–8.

[2] Akeyoshi K, Kanai E. Mechanical properties of tempered glass. In: *Proceedings of the 7th international congress on glass; 1965*. p. 80–5.

[3] Warren P. Fragmentation of thermally strengthened glass. In: *Fractography of glasses and ceramics IV*. The American Ceramic Society.

What is the underlying statistical distribution for the residual stress level and how did you compare it to your photo elastic measurements? Which thickness had which stress level? As the sensitivity of the fragmentation on the stress level is very high (see papers and research given above), a small difference in the stress level will have a very large effect on the fragmentation size. E.g., if you use a 3 mm thick glass and it is subjected to only 35 MPa it will result in a very coarse fracture pattern (strain energy depends linearly on the thickness) but with increasing stress it will change very strongly (strain energy depends to the square on the stress).

According to experimental measurements [4,5], the profile of residual stress through the thickness are approximately parabolic. It is correct that the residual stress depends on the thickness. The thickness is an important parameter. It is for instance known that the minimum residual stress is inversely proportional to the square root of the thickness [3]. In fig. 13 of [31] we compared our numerical predictions with data of Mognato et al. [1] and varied the central tension for various thicknesses.

- [4] Barsom JM. Fracture of tempered glass. J Am Ceram Soc 1968;51(2):75–8.
- [5] Anton J, Aben H. A compact scattered light polariscope for residual stress measurement in glass plates. In: Proceedings of the 8th international glass conference in Tampere, Finland; 2003.

Modeling: How and where did you locate the cohesive zone elements and what was the element's geometry and size? As you have to put the cohesive zone elements in advance, the meshing will influence the fracture pattern ("self fulfilling prophecy"). And the element's size will have a minimum feasible size due to computation time. So the finite element calculation's boundary conditions will influence the results. It is not clear to what extent thus the boundary conditions you put in the finite element code lead to your interpretation of having different breakup processes whereas in reality they are fundamentally the same but only on a different scale. Please discuss and explain better your FE model.

We used 10-node quadratic tetrahedral elements, with average size 0.3 mm, which led to meshes with up to 20 Million degrees of freedom for the largest plates. Cohesive elements are inserted on the fly (extrinsic approach), i.e. they appear when and where the opening stress reaches a critical value that we have set to 70 MPa. The reviewer is correct: unless a fairly complicated adaptive meshing strategy is taken, the crack propagation path will depend of the underlying mesh. In our simulations, the roughness of the fragments, and to a lesser extent the fragments shapes depend on the mesh. However, we have shown in past work that the fragment sizes are fairly insensitive to the underlying tetrahedral mesh. The fragmentation predictions of [31] match well experimental data. We have done also direct comparisons of distribution of fragment sizes using the cohesive element approach with a non-local continuum damage approach. Non-local damage smears the crack thickness over several elements leading to a mesh-independent crack path [6]. Statistics of fragment sizes are in close agreement with these two approaches.

[6] N. Richart and J.F. Molinari, "Implementation of a parallel finite-element library: test case on a non-local continuum damage model", Finite elements in analysis and design, Vol. 100, pp. 41-46, 2015.

Reviewer #3 (Remarks to the Author):

I have read the manuscript at least two times and in my opinion the manuscript is not in a state which can be recommended for publication in Nature Communications. The most significant reason why I make this recommendation is that the authors have not clearly described their motivation for carrying out the work. If they wanted to repeat the findings of the work by Silverman et al. (2012), then they should have stated so. The authors should have noted that in the studies carried out by Silverman et al. (2012), Prince Rupert's drops were made of a lead-crystal glass and they studied fragments collected from disintegration of ~50 drops of head diameters in the range of 10 to 15 mm and overall lengths of ~ 70 to 120 mm.

Although the authors investigated about 2 or 3 Rupert's drops of soda-lime glass of a head diameter of ~ 2.5 mm, a considerable part of their studies was made on the fragments generated by the disintegration of thermally tempered soda-lime glass plates of thicknesses in the range of 4 to 12 mm.

We do not quite understand this comment. Our study goes beyond that of Silverman et al, who measured a limited amount of fragments with an limited resolution. One of our principal results is that the size distribution is *not* a power-law distribution. The difference likely comes from mixing different size distributions of different drops, each with a different size and stress distribution (as Silverman et al. did), but rather from much more precisely looking at the size distribution of a single drop. From our paper: a mm-sized drop explodes into more than 20.000 pieces and we measure its fragment size distribution down to 50 μm . Surprisingly, we find that the size distribution is not power law, but has fragments that follow an exponential size distribution. We believe the apparent power-law of Silverman et al. is due to the mixing of fragments of different drops, and our study and results are thus different and complementary. This is also precisely the motivation of our work : contrary to what for instance Silverman et al. believe, the size distribution is not always power-law. We also explain in detail in our manuscript why this is so. We underlined all this once more in the revised version of the manuscript.

The authors believe that the fragmentation behaviour of a 2-d plate can be likened to the fragmentation behavior of a 3-D Rupert's drop. This is not correct since when a Rupert's drop explodes, many fragments fly out radially, whereas in the case of the fragmentation of a thermally tempered glass plate, the fragments do not fly away, but stay in their pre-disintegration positions (see, for example, Nature 320 (6057) 48-50 (1986)).

We do not believe that the fragmentation behavior is similar, we investigate *whether* the fragmentation behavior is similar between plates and drops. We thus describe both cases in detail in the paper, and show that the fragment size distributions are similar, and similar also to sugar glass broken in a yet different way. The cited Nature paper describes the fracture dynamics, but not the size distribution or difference between 2d and 3d.

Another difference between the fragmentation behaviour of a Rupert's drop made of soda-lime glass and a Rupert's drop made of a lead-crystal glass is that the speed of propagation of disintegration in drops of soda-lime glass is significantly higher than that in Rupert's drops made of a lead-crystal glass. Such a difference may affect the fragment size distribution.

We show in our manuscript that our findings are quite general, and in particular do NOT depend on the type of material. The difference between prince Rupert's drops made of soda lime and lead crystal glass broken in a similar fashion are much smaller than than the breaking of 3d prince Rupert's drops, of glass plates and of sugar glass, all broken in different ways. This shows the robustness of our conclusions. To make this fact clear, we emphasized this once more in the revised version of the manuscript.

My suggestion to the authors is to conduct fragmentations of a much larger number of Rupert's drops made of soda-lime glass and a lead-crystal glass (not plates) and then analyze the results carefully before drawing their conclusions.

Again, based on our observations that we can observe the behavior of the Prince Rupert's drops in the different systems broken in different ways we do not believe that the lead crystal glass would add something to the manuscript. The power-law behavior reported previously is likely to be due to the summing up of a large number of different distributions, each separate one being exponential, and so summing up results from different prince Rupert's drops does not allow for new insights as we present in our manuscript. We have underlined all this in the revised version of the manuscript.

Reviewers' Comments:

Reviewer #1:

None

Reviewer #2:

Remarks to the Author:

The paper was significantly improved and most questions and concerns were answered. Some smaller issues remain for final clarification that might be addressed in the final version of the paper:

- page 4, line 84

I am not so sure why this issue should be too technical for the general audience as the fragment size will be influenced by the hypothesis of a Matern Hardcore Process. In future research, it might be reasonable to assume different levels of the residual stress still present in the fragments and see their influence on the fragmentation sizes based on the assumption of a MHP. As in figure 4(c) you only show the result of one drop (one point), I am not fully sure about this influence and the scattering of the results if using other (or more) drops. Also, I am not sure if this can be seen as the prove for a secondary fragmentation process if a significant part of the energy still might remain in the particles due to the residual stress still present. For future works it is important to improve the methods to measure the stress state in this small particles, I agree. Still, for me it is ok here and only might be mentioned in the Conclusions.

- page 6, line 147

$\alpha = 0.19$ is not really close to 0.5 for me ... something still doesn't fit here. You mention that for glass (as you state at the beginning of line 147) it is typically used a value of 0.5, but you explain further in the sentence that 0.19 fits well with the results? From my point of view you might rewrite the sentence that you see α as an adjustable factor, which is calculated with 0.19 and thus differs from the usually used 0.5. You may want to consider an explanation for this difference?

- Page 7, lines 167 - 170

As you have added that it is an approximate estimation and this is ok for me.

- Page 8 + Conclusions

You still see this as a fully random fracture process. I am still not fully convinced due to the findings in the context of the Matern Hardcore Process mentioned on page 4 and the residual stresses potentially still present in the fragments (see above). Although there is not a prove for the one or the other hypothesis (as the stress level in the particles and also of the original drop could not be measured), maybe you should give a hint to Matern Hardcore processes (page 4, line 84) also in your conclusions.

page 12, line 304

It might be reasonable (but a detail) to state that it is an assumption that the stress distribution is parabolic as you use the SCALP instrument with a reverse fitting of the measured points and a parabola gives good results. This is ok for the tempered plates, their thickness (4 mm to 12 mm) and the level of residual stress you measured. But if quenched very hard, the stress distribution can differ from this and the surface compression stress relatively increases and the neutral lines of stress moves "outward" towards the surfaces. In the case of thin glass tempering (hard quenching) and the drops, this can have an effect.

page 21, line 393

Just for clarification: Did you use the nominal thickness instead of the real measured thickness of the glass (page 12, line 305, 306) in the FE-simulations as a simplification? Did you try to adjust the thickness to the measured thickness in your FE-simulations and/ or get better results in Fig. 4

(c) for the comparison between measurement and calculation for the tempered glass if re-scaling with the thickness? Although figure 4 (c) is very small it seems to me that a reduction in the strain energy (caused by a reduction of the thickness) but keeping the stress level constant might lead to a better the agreement between measured and calculated size.

Reviewer #3:

Remarks to the Author:

I have read authors' response to my comments on the original version of the manuscript as well as the revised version of the manuscript. I am afraid that my opinion of the manuscript has not changed. That is to say, I do not recommend its publication in Nature Communications.

In my original report I asked the authors to clearly state the motivation for doing the work described in the manuscript, but they have not.

The authors do not seem to have understood that the fragmentation behavior of a thermally toughened glass plate cannot be likened to the fragmentation behavior of a Prince Rupert's drop, as the stress distributios in the two cases are quite different from each other. It is for this reason that when a toughened glass plate is shattered, its fragments do not fly outwards from the plate, whereas during the fragmentation of a Prince Rupert's drop, fragments fly outwards (see authors' Fig. 1(a)). In this connection I referred the authors to an article (Nature 320, 48-50 (1986)), but the authors dismissed it by saying it is about dynamics, but so is the fragmentation of a Rupert's drop.

I have found that there is very little physics in the manuscript which convincingly explains why fragments are of different sizes or why there are some fragments as small as 50 μm or even smaller (i.e. fine dust). As regards the fine dust, the authors state in the Conclusions "In normal brittle fragmentation strain energies exceed the values that are needed for equilibrium fragmentation, so that when cracks appear, the breakage process in hierarchic with crack branching at ever smaller length scales to dissipate the excess energy."

The stress in a fragmenting Rupert's drop does not increase, so why should the crack branching length decrease?

Although the title of the manuscript is "Explosive fragmentation of Prince Rupert's drops leads to well-defined fragment sizes", most of the experimental work presented in the manuscript is from two-dimensional plates. Again, in this respect I suggested to the authors to carry out more work on fragmentation of more Rupert's drops, the authors did not do so.

Reviewer #4:

Remarks to the Author:

The authors present convincing evidence, using a combination of clever experimental techniques for different types of glasses backed up by theory and finite element simulations, for the existence of two distinct types of fragmentation processes, hierarchical and random, that lead to power law (scale-free) and exponential (having a characteristic scale) fragment size distributions, respectively. In particular, it is shown that the long-known but still not completely understood type of tempered glass bulbs known as Prince Rupert's drops explode in a random fashion due to their large internal stresses, leading to a fragment size distribution with two exponential regions, with one characteristic length scale for the larger fragments and another for the smaller "dust" that has typically been ignored. The data, spanning 4 orders of magnitude in size, far surpasses that of previous similar studies. I believe that this paper will be of interest both to the broader scientific community as well as to the smaller set of researchers actively studying fragmentation of brittle materials.

The conclusions are well justified, and prior work comprehensively cited, including additional discussion and references added in response to the original reviews. I disagree with Reviewer #3's criticism; particularly in the revised manuscript, the much improved dynamic range of fragment

sizes, and larger statistical sample, is clearly demonstrated, as is the need to question the prior assumption of a single power law distribution.

I did have a minor concern about the comparison of stressed and unstressed plate glass (e.g., Fig. 1), which involves not only different samples, but also different initiation mechanisms - notching an edge vs. dropping on an anvil. A brief (2-3 sentence) discussion of whether the reader should be concerned about this possibly influencing the comparison would be helpful.

Aside from that suggestion, I recommend publication with two mandatory corrections:

1) In the Fig. 2 caption, "shown in (c)" should instead be "shown in (a)".

2) The Fig. 7 legend should define both symbol types. And as somebody with red-green color blindness whom it took some time to ascertain that there were indeed two different data sets, I would also appreciate different color choices (such as the orange and blue in previous figures).

REVIEWER COMMENTS

Reviewer #2 (Remarks to the Author):

The paper was significantly improved and most questions and concerns were answered. Some smaller issues remain for final clarification that might be addressed in the final version of the paper:

- page 4, line 84

I am not so sure why this issue should be too technical for the general audience as the fragment size will be influenced by the hypothesis of a Matern Hardcore Process. In future research, it might be reasonable to assume different levels of the residual stress still present in the fragments and see their influence on the fragmentation sizes based on the assumption of a MHP. As in figure 4(c) you only show the result of one drop (one point), I am not fully sure about this influence and the scattering of the results if using other (or more) drops. Also, I am not sure if this can be seen as the prove for a secondary fragmentation process if a significant part of the energy still might remain in the particles due to the residual stress still present. For future works it is important to improve the methods to measure the stress state in this small particles, I agree. Still, for me it is ok here and only might be mentioned in the Conclusions.

Reply:

Thank you for the positive feedback and constructive comments. We have changed the manuscript accordingly

- We now discuss the Matern hardcore process also in the conclusion, and explicitly discuss the residual stress and their potential effect on the fragmentation sizes
- We added: "Though in this research only one type of Prince Rupert's drop was used, an extended study could focus on including data of multiple drops, measuring stress levels before and after fragmentation as well as using different drop sizes with different levels of tempering. This can be achieved by using different glass types with different thermal expansion coefficients and by changing the frameworking of the glass. The micro-CT measurements of the Prince Rupert's drops fragments are however extremely costly and time-consuming so that perhaps a different technique should be used in such a study."

- page 6, line 147

$\alpha = 0.19$ is not really close to 0.5 for me ... something still doesn't fit here. You mention that for glass (as you state at the beginning of line 147) it is typically used a value of 0.5, but you explain further in the sentence that 0.19 fits well with the results? From my point of view you might rewrite the sentence that you see α as an adjustable factor, which is calculated with 0.19 and thus differs from the usually used 0.5. You may want to consider an explanation for this difference?

Thank you for this comment. After reading again this part of the manuscript, it is apparent the text is a bit ambiguous. It has now be rewritten so that it hopefully leaves no room for multiple interpretations. The model assumes no residual eigenstress after the fragmentation, such that $\alpha = 0.5$. However, the fragments are not stress free, therefore it is common practise (see the reference) to use alpha as an adjustable parameter, which is mostly found to be equal to 0.19, **so also for our results**. Our results therefore match very well with previous work, with $\alpha=0.19$.

- Page 7, lines 167 - 170

As you have added that it is an approximate estimation and this is ok for me.

- Page 8 + Conclusions

You still see this as a fully random fracture process. I am still not fully convinced due to the findings in the context of the Matern Hardcore Process mentioned on page 4 and the residual stresses potentially still present in the fragments (see above). Although there is not a prove for the one or the other hypothesis (as the stress level in the particles and also of the original drop could not be measured), maybe you should give a hint to Matern Hardcore processes (page 4, line 84) also in your conclusions.

We now underline in the conclusion that there is in fact a good indication that there are residual stresses in the remaining fragments (the value of alpha, see above) and conclude that the way to model the fragmentation that is not strictly speaking a random (Poisson) process would be to consider the Matern Hardcore Process.

page 12, line 304

It might be reasonable (but a detail) to state that it is an assumption that the stress distribution is parabolic as you use the SCALP instrument with a reverse fitting of the measured points and a parabola gives good results. This is ok for the tempered plates, their thickness (4 mm to 12 mm) and the level of residual stress you meausred. But if quenched very hard, the stress distribution can differ from this and the surface compression stress relatively increases and the neutral lines of stress moves "outward" towards the surfaces. In the case of thin glass tempering (hard quenching) and the drops, this can have an effect.

Indeed, the assumption that the stress distribution is parabolic will not always be true. For example in chemically tempered glass, though the fracture process shows the same characteristic pattern, its stress distribution is not parabolic. Given the different systems we experimented on, the conclusions should not depend on this in any way. This is in any case a good point, and we have added this to the manuscript.

page 21, line 393

Just for clarification: Did you use the nominal thickness instead of the real measured thickness of the glass (page 12, line 305, 306) in the FE-simulations as a simplification? Did you try to adjust the thickness to the measured thickness in your FE-simulations and/or get better results in Fig. 4 (c) for the comparison between measurement and calculation for the tempered glass if re-scaling with the thickness? Although figure 4 (c) is very small it seems to me that a reduction in the strain energy (caused by a reduction of the thickness) but keeping the stress level constant might lead to a better the agreement between measured and calculated size.

The results of the FE-simulation do not enter figure 4 c), only 4 a), where the results are rescaled such that a difference in thickness does not affect the results. The real tempered glass plates are always slightly thinner than their reported thickness, a neat trick of manufacturers to save material. For the simulations this small detail was not taken into account.

Reviewer #3 (Remarks to the Author):

I have read authors' response to my comments on the original version of the manuscript as well as the revised version of the manuscript. I am afraid that my opinion of the manuscript has not changed. That is to say, I do not recommend its publication in Nature Communications.

In my original report I asked the authors to clearly state the motivation for doing the work described in the manuscript, but they have not.

The authors do not seem to have understood that the fragmentation behavior of a thermally toughened glass plate cannot be likened to the fragmentation behavior of a Prince Rupert's drop, as the stress distributios in the two cases are quite different from each other. It is for this reason that when a toughened glass plate is shattered, its fragments do not fly outwards from the plate, whereas during the fragmentation of a Prince Rupert's drop, fragments fly outwards (see authors' Fig. 1(a)). In this connection I referred the authors to an article (Nature 320, 48-50 (1986)), but the authors dismissed it by saying it is about dynamics, but so is the fragmentation of a Rupert's drop.

I have found that there is very little physics in the manuscript which convincingly explains why fragments are of different sizes or why there are some fragments as small as 50 μm or even smaller (i.e. fine dust). As regards the fine dust, the authors state in the Conclusions "In normal brittle fragmentation strain energies exceed the values that are needed for equilibrium fragmentation, so that when cracks appear, the breakage process in hierarchic with crack branching at ever smaller length scales to dissipate the excess energy."

The stress in a fragmenting Rupert's drop does not increase, so why should the crack branching length decrease?

Although the title of the manuscript is "Explosive fragmentation of Prince Rupert's drops

leads to well-defined fragment sizes", most of the experimental work presented in the manuscript is from two-dimensional plates. Again, in this respect I suggested to the authors to carry out more work on fragmentation of more Rupert's drops, the authors did not do so.

Thank you for the comments on our manuscript.

It is unclear to us why the motivation for our work remains unclear. In the previous round we in detail responded to all of your comments in detail, especially the comments about the work of Silverman et al. in relation to our own results, and even included this in a graph in the manuscript. As also noted by the other reviewer (#4 in this document), our results greatly surpasses that of any previous similar studies and as stated by another reviewer: *"The experimental methods developed and applied by the authors to explore this issue are innovative, quite original and effectively address the issue of interest."*

Unless the reviewer can express in a more precise matter what is lacking, it is not clear to us what should be further motivated.

We are of course aware of the different stress profiles in the materials used. But this is precisely one of the interesting conclusions of our research. Though systems can be quite different, their breakup process can be the same. That is why power law distributed fragments appear in a great number of experiments, despite the very different materials used, while in some cases the fragment size distribution is exponential. We tried to demonstrate this by showing that one can break the same material in two different ways, one giving exponentially sized fragments, while the other results in power law sized fragments.

Regarding the cited sentence from the manuscript, this sentence is not referring to the 'dust' particles, but to the breakup process. The fine dust is not even mentioned in the conclusion. The branching of cracks describes the difference between the two breakup processes, leading to fundamentally different fragment size distributions, i.e. scale free vs not scale free. We hope this explains the reviewer's question.

Indeed the manuscript contains many other experiments besides the fragmentation of Prince Rupert's drops. This is to support the presented explanations and conclusions, that with only results from measured fragmented sizes of Prince Rupert's drops would be quite audacious.

We strongly encourage the reviewer to be more specific in his/her comments, other than requesting more experiments, as one can always do more experiments. As previously stated and noticed by the reviewer himself/herself, our manuscript contains many experiments on many different systems. Other reviewers also don't see the need for more experiments, so a more detailed response would be required for answering any of

the reviewer's concerns.

Reviewer #4 (Remarks to the Author):

The authors present convincing evidence, using a combination of clever experimental techniques for different types of glasses backed up by theory and finite element simulations, for the existence of two distinct types of fragmentation processes, hierarchical and random, that lead to power law (scale-free) and exponential (having a characteristic scale) fragment size distributions, respectively. In particular, it is shown that the long-known but still not completely understood type of tempered glass bulbs known as Prince Rupert's drops explode in a random fashion due to their large internal stresses, leading to a fragment size distribution with two exponential regions, with one characteristic length scale for the larger fragments and another for the smaller "dust" that has typically been ignored. The data, spanning 4 orders of magnitude in size, far surpasses that of previous similar studies. I believe that this paper will be of interest both to the broader scientific community as well as to the smaller set of researchers actively studying fragmentation of brittle materials.

The conclusions are well justified, and prior work comprehensively cited, including additional discussion and references added in response to the original reviews. I disagree with Reviewer #3's criticism; particularly in the revised manuscript, the much improved dynamic range of fragment sizes, and larger statistical sample, is clearly demonstrated, as is the need to question the prior assumption of a single power law distribution.

We would like to thank the reviewer for these nice words about our work and the acknowledgement of the relevance of our results. We agree that Reviewer #3's, critic to the relevance of our work is somewhat confusing, as we believe our results are far better than any previous work and our conclusions and experiments are very different as well.

I did have a minor concern about the comparison of stressed and unstressed plate glass (e.g., Fig. 1), which involves not only different samples, but also different initiation mechanisms - notching an edge vs. dropping on an anvil. A brief (2-3 sentence) discussion of whether the reader should be concerned about this possibly influencing the comparison would be helpful.

Thank you for this comment. The different methods and materials can of course raise concern. Therefore, we also included experiments on sugar glass where we used the same material as well as shape, where only the breakup method differs. One of the main points is of course that, for the stressed plates it doesn't really matter how you initiate

the first cracks, the outcome is always the same (i.e. the same distribution, not the same pattern of course). Similarly, for unstressed glass plates, independent of the breakup method, the fragment size distribution will be power law. From an experimental point of view, when the plate only breaks in a few pieces, it is very hard to construct a size distribution. Hence the different methods. We included a few sentences to hopefully satisfy the concerned reader.

Aside from that suggestion, I recommend publication with two mandatory corrections:

- 1) In the Fig. 2 caption, "shown in (c)" should instead be "shown in (a)".
- 2) The Fig. 7 legend should define both symbol types. And as somebody with red-green color blindness whom it took some time to ascertain that there were indeed two different data sets, I would also appreciate different color choices (such as the orange and blue in previous figures).

Thank you for these corrections, we made changes accordingly. We indeed want everybody to be able to look at our data.

Reviewers' Comments:

Reviewer #2:

Remarks to the Author:

From my view, the authors addressed the questions raised and the paper is fine for publication.